# Time efficiency as a mediator between institutional support and higher education student engagement during e-learning

**Tarik Abdulkrem Alwerthan** *

Department of Educational Leadership and Policies, College of Education, Taif University, Taif, Kingdom of Saudia Arabia

* tawerthan@tu.edu.sa

## Abstract

This study examined the relationship between institutional support and student engagement in e-learning, with time efficiency as a potential mediator among Saudi university students. This study employed a cross-sectional, questionnaire-based research design. A sample of 752 Saudi university students from different provinces in the Kingdom of Saudi Arabia completed an online questionnaire. The results of the measurement model showed good reliability and validity for all constructs. The results of the structural model and hypothesis testing showed that this study partially supported the hypotheses. Notably, there was a significant positive relationship between student support during e-learning and student engagement. In addition, student support was found to be a predictor of student's time efficiency (short-term planning, long-term planning, time attitude). Furthermore, the results revealed that long-term planning and time attitude, both components of time efficiency, act as mediators between support provided to students by a higher education (HE) institution during e-learning and their level of engagement. The findings shed light on the underlying mechanisms that explain students' e-learning engagement.

## Introduction

E-learning in higher education has existed over several decades but has undergone multiple radical changes in the past five years as a result of the COVID-19 pandemic buoyed by prior [1–3]. One of the most significant and far-reaching of those changes is the implementation of e-learning. E-learning, a form of distance education, involves courses delivered through virtual classrooms, enabling students to participate in and gain knowledge of required skills regardless of time and place [4,5]. Pre-COVID-19, only a few learning institutions (early adopters regarded as enthusiasts and visionaries) implemented online or blended learning methods [6,7]. When the pandemic hit, the pandemic 1.6 billion students and youth worldwide, prompting a switch from offline to online classes across 191 countries in the world (98% of the global student population (The United Nations Educational, Scientific and Cultural Organization [UNESCO], [8]). E-learning not only makes it easier for individuals to continue their

Zenodo at https://doi.org/10.5281/zenodo.13952134.

**Funding:** The authors would like to acknowledge Deanship of Graduate Studies and Scientific Research, Taif University for funding this work. The funder had no role in study design, data collection and analysis, decision to publish, or preparation of the manuscript.

**Competing interests:** The author has declared that no competing interests exist

learning journey despite geographical constraints [9,10], but is also a sustainable form of educational development for those same reasons. According to UNESCO [11] education for sustainable development is a lifelong learning process and an integral part of quality education. It enhances the cognitive, socio-emotional, and behavioral dimensions of learning and encompasses learning content and outcomes, pedagogy, and the learning environment itself.

In Saudi Arabia, the rapid transition to e-learning saw the Saudi Arabian government announce an immediate shift to online learning on March 8, 2020, for all educational institutions, including universities, colleges, and K-12 schools [12]. This transition, facilitated by pre-existing e-learning infrastructure the Saudi Ministry of Education had already provided to public universities to support virtual classrooms and online learning, led to all universities in Saudi Arabia utilizing the Blackboard platform by 2022 [12,13]. With 61 universities and colleges involved (with more than 1.4 million students and 76,000 faculty members), over 35 million learning sessions handling more than 69 million logins, facilitating over 300,000 courses, and administering more than 25 million assessments and tests were recorded [13]. This rapid and extensive adoption of e-learning in Saudi Arabia presents a unique context for studying the dynamics of student support, engagement, and time efficiency in online educational environments.

Studies established that effective, sustained use of e-learning would largely be actualized through student support through staff and faculty [14]. Appreciating the role of student support in e-learning, education institutions have begun to prioritize student empowerment or support beyond student engagement [15–18]. Studies have associated student empowerment or support predicting student attainment of educational goals [16,17,19] and as an indicator for student-engagement [20,21]. Studies have revealed that high levels of student engagement correlate positively with higher grades [22], successful learning [23–25], academic resilience, attendance [26], school completion and retention [27]. Thus, it is essential for institutions to increase student engagement so that e-learning can be encouraged. The level of student support received during e-learning can also determine if the student has a successful experience or not [28]. However, student support has been perceived as one of the most challenging aspects of e-learning in HE institutions, which indicates that its connection to student engagement needs to be studied further [20,21].

Student time-efficiency, another variable of note to student engagement in e-learning, is significant as it plays a major role for students in task completion and content mastery [29–31]. In a study conducted by Adams and Blair [32] data collected from 289 students from the Department of Electrical and Computer Engineering of Trinidad and Tobago revealed 12 factors that influenced students' performance in HE. Specifically, the results of their study indicated that time efficiency is the most significant factor influencing student performance. Hence, time efficiency can potentially mediate the relationship between student support and student engagement in an e-learning environment. While those who poorly use their time are less likely to succeed, individuals who effectively use their time are more likely to perform better, a result of focusing on purposeful activities that make fixed goals achievable [33–38]. Since an e-learning environment can be characterized by distractions (like social media use, texting, television, and family) [37,39,40], including time efficiency to investigate the association between student support and student engagement is necessary.

As such, this study sought to examine the relationship between student engagement and their perceptions toward received social support during e-learning, using the framework of engagement theory [41] and self-determination theory of motivation [42,43] Additionally, the current study aims to investigate the role of students' efficient use of time as a mediator and explain links between student-perceived support and engagement using the time management framework.

The study is guided by three research questions:

1. What is the relationship between student support during e-learning and student engagement?

2. What is the relationship between student support during e-learning and student time efficiency?

3. How does time efficiency mediate the relationship between student support during e-learning and student engagement?

## Literature review and hypothesis

### Time-efficiency during e-learning

Time efficiency has been studied across various domains like education, business management, psychology, and organizational behavior. However, studying time efficiency as a mediator between student support and student engagement has not been directly covered. Educational time efficiency scholars propose that students' time efficiency (involving planning, prioritization, goal-setting, and time allocation) translates to full engagement and suggests perceived student self-regulation [44]. Generally, self-regulation has been associated with predicting positive outcomes like increased student performance in traditional learning environments and e-learning environments, e.g., [45]. Hence, students' time efficiency (suggesting perceived student self-regulation) can significantly impact the quality of the e-learning environment [33,46,47]. Despite its acknowledged relevance, a lack of consistency in conceptual understanding and measurement of time efficiency is observed in time management intervention studies [32,48,49]. Time-efficiency scholars also tend to divide it into dimensions such as long-term planning, short-term planning, and time-attitude [17,33,37,50]. However, regardless of how it is measured, research has shown that there is a positive association between time-efficiency and outcomes such as student engagement and positive attitudes towards learning [17,32,37,51]. For instance, in a systematic review that utilized a meta-analysis, Broadbent and Poon [52] found a positive correlation between students' time efficiency during e-learning and academic outcomes.

In daily living, individuals tend to show a desire to organize themselves as it is key to controlling self-conduct and achieving intended objectives [50,53,54] suggesting that time efficiency is a crucial e-learning behavior. For example, in an experimental study conducted on 118 university students, Trentepohl and others [55] concluded that an intervention focused on time-efficiency practice increased students' time-efficiency skills more than an intervention focused on time-efficiency knowledge. The students' performance was also enhanced. Moreover, a separate study found that while self-regulation during e-learning had a significant effect on time efficiency, stress and academic procrastination were seen to have no significant impact on time efficiency [33]. In another study involving 75 Engineering students, Adams and Blair [32] highlighted that students' time efficiency predicted having high GPAs and higher grades in their program. Similarly, in a study involving 153 masters and doctoral students, Gupta and Chitkara [37] noted that the level of time efficiency (long-term and short-term) significantly predicted students' academic performances. Furthermore, in a quantitative study involving second-year students enrolled in a virtual course at a Turkish university, Er [30] found significant differences in the degree of student engagement with different course components (video recording, in-class exercises, quizzes, lab assignments) based on their self-reported effective use of time.

### E-learning student-engagement and engagement theory ET

Engagement theory advances that interactive and meaningful tasks can significantly improve engagement levels [41]. Engagement theory, advanced by Kearsley and Shneiderman [41], provides a reliable framework for interpreting how interaction and meaningful tasks in technology-based learning environments can enhance student engagement. The theory posits that students connect more deeply with learning when they are involved in collaborative and project-based activities, which foster authentic, creative, and interactive learning experiences [41]. This theory is especially relevant in e-learning contexts, where the use of technology, such as online tools and mobile devices, can support the intellectual, social, and behavioral forms of engagement [56,57]. Engagement theory promotes active participation and meaningful interactions, offering a foundation for assessing how students' use of technology can improve focus, motivation, and overall learning outcomes.

Student engagement, an important topic in traditional, online, and blended learning, is the level of focus, curiosity, interest, and enthusiasm students demonstrate toward their online education, which can influence their learning outcomes [36,41,51]. Several measures have been advanced based on conceptual and theoretical backgrounds to evaluate student engagement during e-learning, and each one has its own limitations and strengths. According to Finn and Zimmer [58], the concept of "student engagement" was used in the 1980s as a tool to reduce student dropout rates, boredom, and alienation. Scholars have provided different models to measure student engagement in learning contexts with Soffer and Cohen [59]. For example, suggesting that students' active engagement in academic courses can be measured by looking at three angles: engagement with materials, engagement between students and faculty members, and students' performance. In a study that aimed to develop an instrument to measure student engagement in e-learning environments, Lee et al [60] built a measure comprising six factors (community support, peer collaboration, interaction with faculty members, cognitive problem-solving, psychological motivation, and learning management). Consequently, after reviewing related studies, Chiu [61] gathered data from 737 students who were enrolled in Korean universities and measured student engagement by adopting four dimensions: behavioral engagement, emotional engagement [25], cognitive engagement [62], agentic engagement [63].

Several studies have highlighted the positive impacts of student-engagement [59,60]. Redmond et al. [64] advanced that college students' engagement was perceived as a significant benchmark and indicator of educational quality for student experiences. In a study involving 646 nurse students, Soffer and Cohen [59] noted that students who were engaged were more likely to complete courses. Also, in a longitudinal survey study involving 851 HE students in Finland, Upadyaya and Salmela-Aro [22] found that student engagement was a predictor of satisfaction with a decided educational journey, higher grades, and positive educational transitions. As such, the current study advances that students' engagement in e-learning is a result of receiving academic institution support and student level of time efficiency. In a survey of 288 Malaysian students enrolled in undergraduate business and management programs, Putit et al. [65] revealed that student's behavioral control and attitude can predict the level of student learning engagement. Similarly, a cross-sectional study conducted by Liu et al. [66] found that perceived support by HE students was a predictor of their level of academic engagement.

### Motivation and student support in e-learning environment

Motivation has been utilized in several contexts, including educational settings. This review reveals several aspects of student support in the e-learning environment, beginning with academic support [67] which encompasses assistance provided to enhance learning and academic

performance. Whereas social support involves encouragement from peers, faculty, and staff, it has [68,69] emotional support aids in managing stress and other emotional challenges [70]. While technical and digital support includes IT help and software guidance [61,71,72], institutional support comprises service, technology, and digital support [73]. Teacher support, including personalized feedback and guidance, has also been associated with influencing student engagement [74,75]. These various forms of support collectively contribute to fulfilling students' basic needs for competence, autonomy, and relatedness, as posited by self-determination theory [42,43], potentially leading to better time efficiency and higher engagement levels in e-learning environments.

Several studies have shown that student support can positively impact students in e-learning environments [73,76–79]. For example, in a qualitative study involving ten college students in Vietnam, Pham et al. [78] found that student support was crucial to alleviating some of the challenges students faced in e-learning environments. Support from authority figures like teachers was also important for enhancing student engagement. In a study that collected data from 1136 Chinese higher vocational students, Xu et al. [74] showed that perceived teacher support also influenced student engagement. Additionally, Fall and Roberts [75] found that middle school students' perceptions of their teachers' support positively predicted their level of engagement. This highlights the influence that authority figures and peers might have on engagement. As such, faculty members and staff can serve as vital agents of student support during e-learning.

From a self-determination theory perspective, student support enhances motivation by focusing on psychological need satisfaction (as preeminent behavioral determinants) and fulfilling students' basic needs for competence, autonomy, and relatedness, leading to better time efficiency and higher engagement levels [42,43]. In a quantitative study involving 1,201 students enrolled in grades 8 and 9 in middle schools in China, Chiu [61] found that proper support during e-learning significantly predicted high levels of student engagement. In a study conducted over one academic year involving 975 first-year Japanese university students studying English as a foreign language, Fryer and Bovee [80] found that student support was linked to a broad range of direct effects on students' motivations for e-learning. In other words, an environment that supports students' basic psychological needs enhances their levels of engagement in beneficial behaviors. In a quantitative study involving 1,512 university Chinese students during COVID-19, She et al. [70] showed that education support positively influenced students' perceptions of e-learning usefulness and ease of use, while emotional support had a positive influence only on perceived use. In another study, Sokman et al. [81] found that the student-to-instructor and the student-to-content interaction were essential elements in an e-learning environment, signaling the influence that others can have on one's perceptions of something new, like e-learning. This support then impacts students' behavior in long-term, short-term, and time-based attitudes during e-learning. As a result of the support that other people can have on the perceptions of changes within an environment, Rotar [68,69,82] advance that student support among peers would act as a significant predictor of student engagement, success, and overcoming potential challenges in e-learning environments.

Students also routinely require academic and technical support in e-learning environments. Essel et al. [71,83] determined that every student needs some form of assistance to use technology and learn effectively in an e-learning environment, with technical support significantly predicting students' success in e-learning. In a cross-sectional survey involving 240 Indian college students, Kakada et al. [73] noted that university support (service support, technology support, social support, & academic support) was positively associated with student satisfaction. Chiu [61] also found that digital support coupled with competence and autonomy significantly predicted cognitive and emotional engagement, translating to enhanced digital literacy

and self-regulated learning. Subsequently, the current study proposes that the level of perceived support by students can be an indicator of their engagement and time efficiency.

This study fills several key gaps in the existing e-learning literature by presenting a holistic approach to understanding student success in online educational environments. First, it investigates time efficiency as a significant mediator between student assistance and student engagement in e-learning situations, a previously untapped topic. This broadens engagement theory to include time efficiency as a fundamental component in online learning, where time management may be even more important than in traditional settings. Second, the study provides an enhanced model that incorporates institutional support, time efficiency, and engagement, providing fresh insights into how these variables interact in online education. This addresses a gap in research on how these aspects interact in e-learning environments. Thirdly, by using the Self-Determination Theory in this context, this study provides an enhanced understanding of how student support affects time efficiency and engagement, which may be mediated by satisfying basic psychological needs. This method fills gaps in existing research and provides essential theoretical and practical insights into improving student results in e-learning. By addressing these interconnected aspects, this study contributes to a more comprehensive understanding of the factors influencing student success in online educational environments, giving institutions insights into how to more effectively foster student engagement and improve time management skills through targeted support.

**Study hypotheses.**   Based on what the previous sections highlighted, this study hypothesized the following:

*H1*: *Higher levels of all three forms of time efficiency predict higher levels of student engagement* (see Fig 1).

*H2*: *Higher levels of student support predict higher levels of every form of time efficiency* (see Fig 2).

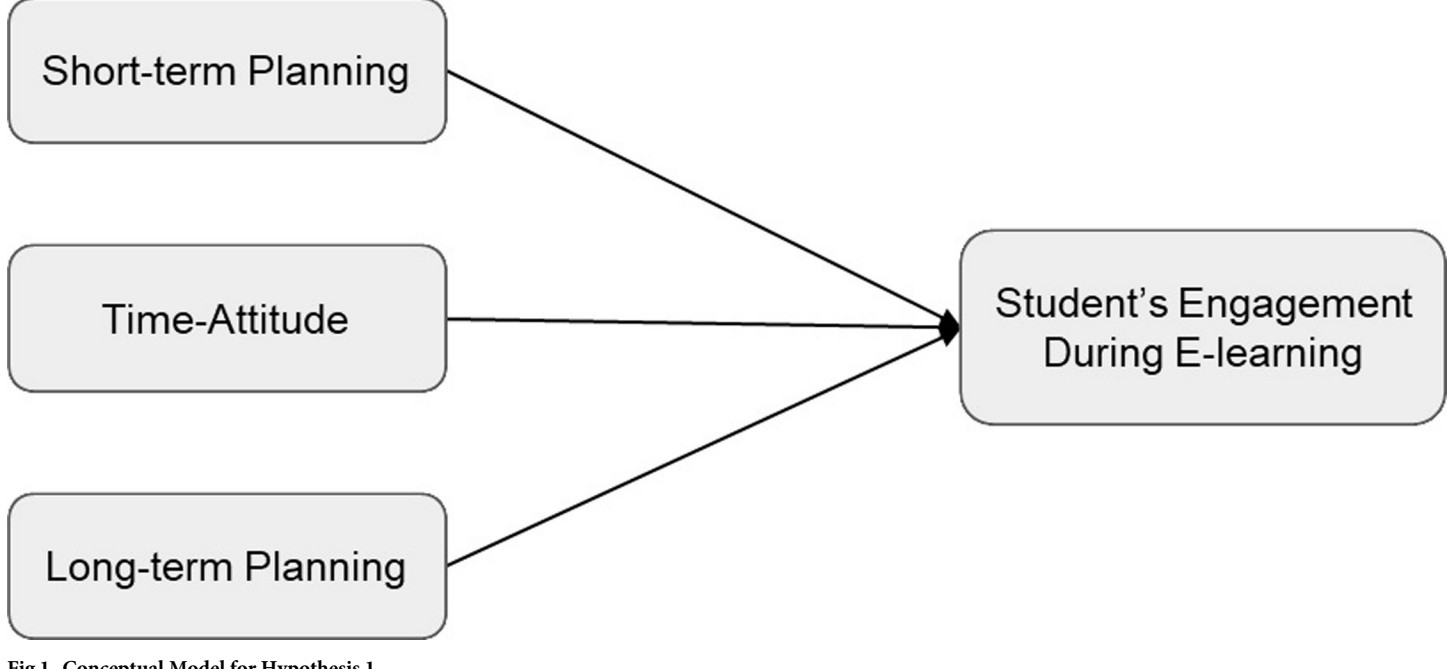

**Fig 1. Conceptual Model for Hypothesis 1.**

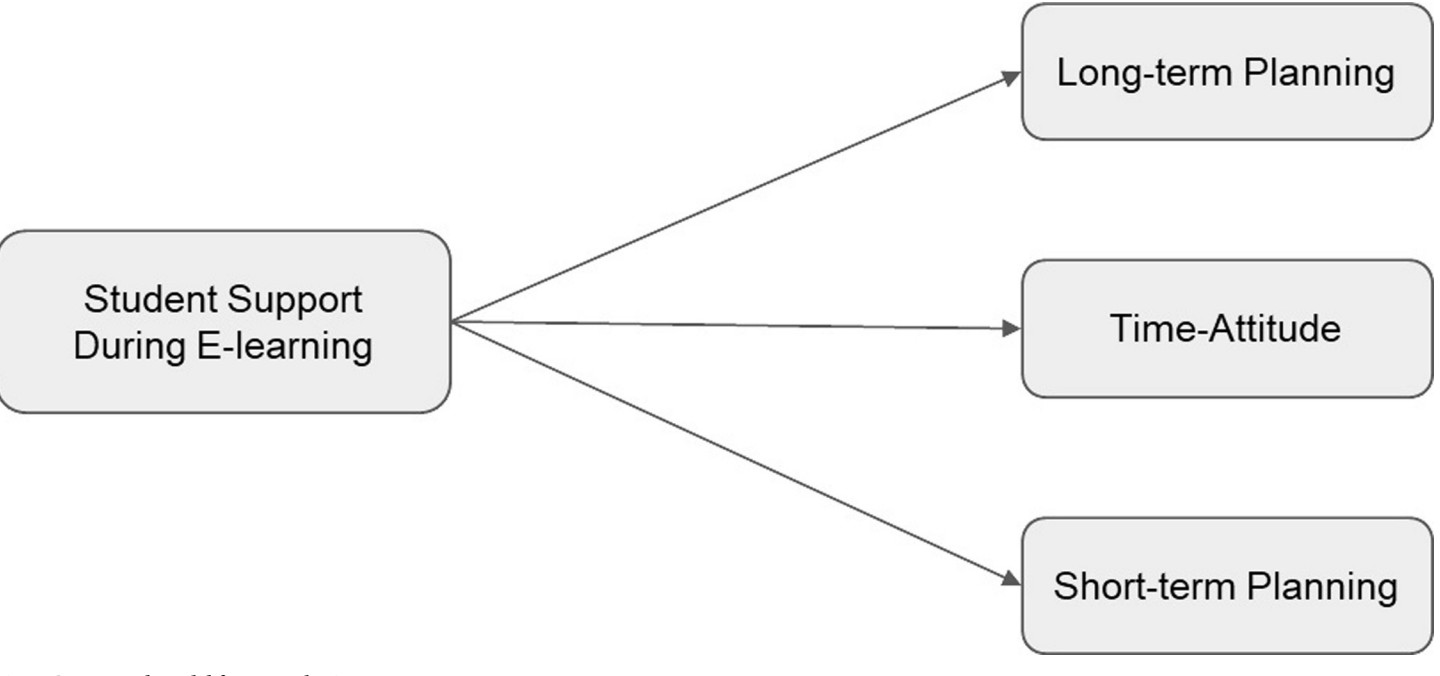

**Fig 2. Conceptual Model for Hypothesis 2.**

*H3*: *Higher levels of student support predict higher levels of all three forms of time efficiency, which in turn predict higher levels of student engagement* (see Fig 3).

## Methods

### Procedure and participants

This study employed a convenience sampling strategy to recruit student participants from Saudi universities exposed to e-learning environments during the COVID-19 pandemic. As noted by Andrade [84], convenience sampling is characterized by the researchers drawing participants from a source that is conveniently accessible to the researchers. While this approach allowed for efficient data collection, cost-effectiveness, and ease of implementation [85],

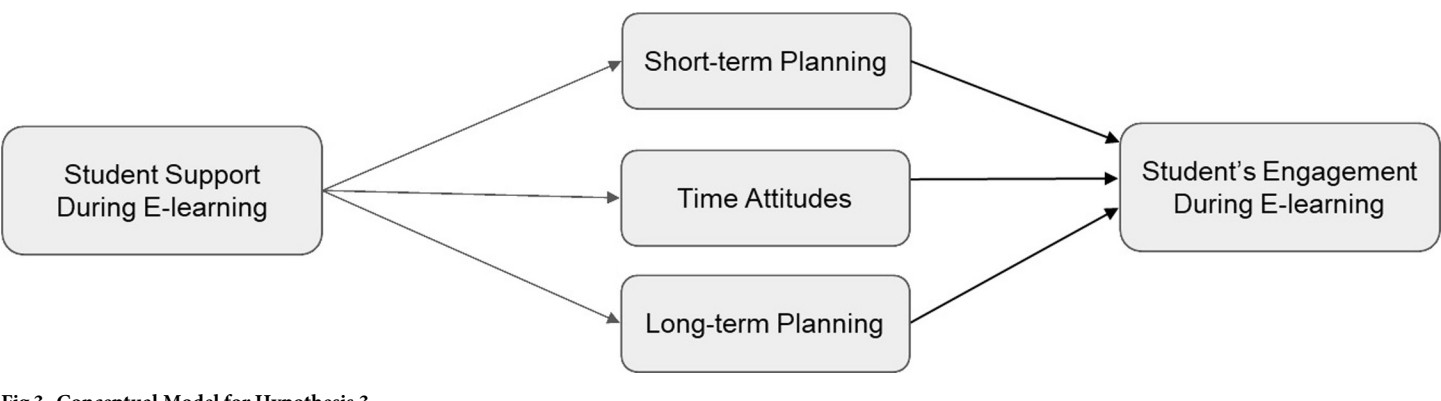

**Fig 3. Conceptual Model for Hypothesis 3.**

Andrade [84] notes that its primary limitation lies in compromised external validity. While studies using convenience samples could guarantee high internal validity if methodologically accurate, the findings may have limited generalizability beyond the specific population from which the sample was drawn.

This study's student participants were recruited from Saudi universities exposed to e-learning environments between September 20[th], 2022, and November 11th, 2023, following the onset of the COVID-19 pandemic. There were 75 respondents who fully completed the survey: 57.06% female, 41.6% male. The respondents were enrolled in the targeted universities to obtain bachelor's degrees (100%) but drawn from different disciplines (Humanities 149:19.86%, Sciences 149:19.86%, Management 150: 20%, Health 168: 22.4%, and Engineering 136: 18.13%). The study was approved by an institutional review board (IRB: Taif University's ethics approval reference [HAO-02-T-105: 45–023]) and required completing a 12–15 minute anonymous electronic survey, which was designed to assess student support, student time efficiency, and student engagement from students' perceptions. The initial section of the online survey provided an introduction detailing participants' rights and seeking their consent. Respondents were sensitized to their right to participate voluntarily and withdrew their participation with no negative consequences.

There were direct or indirect participant identifying items on the questionnaire to guarantee the confidentiality of their responses. Subsequently, participants were prompted to confirm whether they accepted or declined to participate in the study. Participant consent was obtained by providing the following question at the end of the introductory online page: "Do you want to participate in this study?". If students chose "yes," they would voluntarily complete the questionnaire. If a student's choice was "No," the following statement appeared to them: "Thank you for taking time to read about the current study in this introductory online page," implying respect for their choice not to participate.

## Measures

Three instruments were used to gather the data for this study: a general student support instrument, a student time-efficiency instrument, and a student engagement instrument. A 5-point response scale ("Not at all, a little, somewhat, quite a bit, very much extremely") was used for the instruments' items. The *student support instrument* is a method of assessing the support received by students to solve any technical issue, academic challenge, or administrative matter from the academic institution, faculty member, staff, or technical experts (e.g., *The e-courses' plans are clear and available*). The responses indicated a high internal consistency in the current sample (Cronbach's α = 0.929). The *student time-efficiency instrument* is a method of assessing to what extent students engage in behavior that reflects their levels of wise use of time. This instrument has three subscales, which include [50,86,87]: student attitude toward time (e.g., Cronbach's α = 0.731), student short-range planning (e.g., *I write goals that I work to achieve daily*; Cronbach's α = 0.922), and student long-range planning (e.g. *I continue to carry out unprofitable routines or activities during e-learning*; Cronbach's α = 0.777). The *student-engagement instrument* is a method of assessing the extent of students' active involvement during e-learning activities. This instrument contains four subscales [61]: behavioral engagement (e.g., *When I'm in online learning, I participate in synchronous and asynchronous discussions*; α = .944), cognitive engagement (e.g., *I think about different ways to solve a problem;* α = .901), emotional engagement (e.g., *When we work on something in online learning, I feel interested;* α = .949), and agentic engagement (e.g., *During online learning, I express my preferences and opinions*; α = .917).

# Results

## Data analysis

The data analysis phase of this study encompassed evaluating responses gathered from 752 students across three distinct universities (Taif University, Prince Sattam University, and Shaqra University). Univariate and multivariate normality tests using IBM SPSS Statistics 29.0 were conducted to uphold the validity of statistical modeling techniques. The results from the Shapiro-Wilk and Kolmogorov-Smirnov tests indicated deviations from normal distribution, with some variables displaying skewness and kurtosis values beyond conventional thresholds. A complementary multivariate normality evaluation using AMOS confirmed notable deviations from normality, thereby challenging the assumptions integral to covariance-based SEM methods traditionally executed in AMOS. Subsequently, the analysis was steered towards Partial Least Squares SEM (PLS-SEM), utilizing SmartPLS software. This approach is notably adept at managing datasets that do not conform to standard distribution assumptions and is well-suited for exploratory analysis and theory development. PLS-SEM's capability to accommodate complex models with numerous constructs and indicators made it particularly suitable for the dataset in question. The rationale for this shift in methodology was twofold. Firstly, the reliance on covariance-based SEM on multivariate normality was incongruent with the observed data characteristics. Continued application of AMOS could lead to misestimated parameters and a misconstrued evaluation of the model's fit. Secondly, the preference for PLS-SEM within SmartPLS due to its leniency towards non-normality ensures that the predictive models derived are robust and reliable.

The succeeding segments validate measurement models, assess the structural model to test hypotheses, interpret results in context, discuss implications and limitations, and refine constructs iteratively to ensure validity and reliability in the PLS-SEM model. The objective is to ensure both convergent and discriminant validity and the reliability of the constructs. This is in line with the suggestions made by Hair et al. [88] that constructs should exhibit high internal consistency and be distinctive from each other.

# Results

## Demographic data

Table 1 highlights the demographic characteristics of the participants in the study, including their age range, gender distribution, university affiliation, academic specialty, and year of study. The age range of university students who answered the study was 17 to 54 years old. The vast majority of responses were between the ages of 18 and 22. A lower proportion of students were in their late twenties to early forties, with a few responders older than 40. According to the data, the most common age groups were 19 and 20, followed by 21 and 22. There were also distinct groups of students aged 30, 37, 38, and 40, with fewer responses among those aged 45 and up. The study included 428 men (57.8%) and 312 women (42.2%). Respondents were evenly dispersed among three universities: Taif (33.4%), Prince Sattam (33.2%), and Shaqra (33.4%). Regarding specialty, 19.8% of participants were from Humanities, 19.8% from Sciences, 19.9% from Management, and 22.3% from Healthcare. According to their year of study, 26.5% were sophomores, 41.1% were juniors, and 25.4% were seniors.

## Interpretation of findings

The initial analysis revealed that several items across the constructs of short-term planning, long-term planning, time attitudes, and student engagement suffered from issues of convergent and discriminant validity. This is consistent with the recommendations by Hair et al. [89]

**Table 1. Sample description (N = 752).**

| Category | Sub-category | Frequency | Percentage |
|---|---|---|---|
| Gender | Male | 428 | 56.91% |
| | Female | 312 | 41.49% |
| | Missing Data | 12 | 1.60% |
| University | Taif University | 251 | 33.40% |
| | Prince Sattam University | 250 | 33.20% |
| | Shaqra University | 251 | 33.40% |
| | Missing Data | 0 | 0% |
| Specialty | Humanity | 149 | 19.81% |
| | Sciences | 149 | 19.81% |
| | Management | 150 | 19.95% |
| | Health | 168 | 22.34% |
| | Missing Data | 136 | 18.09% |
| Year of Study | Sophomore | 199 | 26.46% |
| | Junior | 309 | 41.09% |
| | Senior | 191 | 25.40% |
| | Missing Data | 53 | 7.05% |

*Note*: Missing data refers to participants who did not provide a response for the corresponding category.

and Dijkstra and Henseler [90] that constructs should ideally explain more than 50% of the variance of its items (as indicated by an AVE greater than 0.5), and that the square root of a construct's AVE should be larger than its correlations with other constructs to establish discriminant validity. In a subsequent process of elimination, several indicators across the constructs 'short-term planning,' 'long-term planning,' 'time-attitudes,' and 'student-engagement' exhibiting low factor loadings were identified as problematic and subsequently removed from the model consistent with previous studies in the field [17,29,33,37,61,91]. This step aligns with the benchmark provided by Hair et al. [88] that loadings above 0.708 are recommended. Furthermore, items with VIF values higher than the threshold of 5, which contributed to multicollinearity, were also excluded from the model, as suggested by Becker et al. [92]. These actions were taken to enhance the model's integrity and to meet the rigorous standards for construct reliability and validity as posited by Drolet and Morrison [93].

**Measurement model assessment.** As summarized in **Table 2**, the measurement model was assessed using several priority validity and reliability factors. Internal consistency was measured using Cronbach's alpha values, which ranged from 0.798 for 'Long-term Planning' to 0.935 for 'Student Engagement,' all of which were over the proposed cutoff of 0.7 [94]. This suggests that all structures have a high level of internal consistency. The reliability of the constructs was further validated by composite reliability measures, which included both rho_a and rho_c. All of the rho_a and rho_c values were substantially over the acceptable threshold,

**Table 2. Construct reliability and validity.**

| | Cronbach's alpha | Composite reliability (rho_a) | Composite reliability (rho_c) | Average variance extracted (AVE) |
|---|---|---|---|---|
| Long-term Planning | 0.798 | 0.822 | 0.806 | 0.678 |
| Student Engagement | 0.935 | 0.94 | 0.936 | 0.649 |
| Student Support | 0.929 | 0.933 | 0.929 | 0.622 |
| Time Attitudes | 0.804 | 0.819 | 0.809 | 0.681 |

ranging from 0.819 to 0.940 and 0.806 to 0.936, respectively. The Average Variance Extracted (AVE) was used to evaluate convergent validity. Each construct showed AVE values over the 0.5 cutoff [95], with 'Student Support' showing AVE values of 0.622 and 'Time Attitudes' showing AVE values of 0.681. This implies that the items capture the variance of their respective constructs in an appropriate manner. Consistent with Hair et al. [94], the findings point to the measurement model's high convergent validity and reliability, which offer a reasonable basis for additional structural model research.

As summarized in **Table 3**, The Heterotrait-Monotrait (HTMT) ratio of correlations was used to assess the measurement model's discriminant validity and evaluate the uniqueness of its constructs, as Henseler et al. [96] advanced. The discriminant validity of the model was found to be adequate, with all HTMT values falling below the conservative criterion of 0.90. 'Long-term Planning' and 'Student Engagement' had the highest HTMT score, 0.88, which was closely followed by 'Student Support' and 'Student Engagement,' 0.841. Other noteworthy HTMT values were 0.778 between 'Long-term Planning' and 'Student Support' and 0.81 between 'Long-term Planning' and 'Short-term Planning.' Of all the constructs, 'Time Attitudes' had the lowest HTMT values, ranging from 0.368 (with 'Short-term Planning') to 0.592 (with 'Student Engagement'), suggesting that 'Time Attitudes' is significantly different from the other model constructs.

**Structural model assessment.** The structural model was assessed using various critical metrics, including multicollinearity, explanatory power, predictive relevance, and out-of-sample predictive power. In assessing the potential for multicollinearity within the structural model, the reported VIF values ranged from a low of 1.567 for the 'CE25' variable to a high of 4.648 for 'AE28'. All values comfortably resided below the critical threshold of 5, as suggested by Becker et al. [92], the risk of multicollinearity compromising the reliability of the structural model estimates is mitigated. The explanatory power of the model, captured by the $R^2$ values, reflects the variance in the dependent constructs accounted for by the exogenous variables. 'Student engagement' emerged with a moderate $R^2$ value of 0.674, suggesting the model explains a substantial amount of the variance within this construct. In contrast, 'long-term planning,' 'short-term planning,' and 'time-attitudes' reported lower $R^2$ values of 0.464, 0.428, and 0.183, respectively, indicating a more modest explanatory capacity. These findings prompt consideration of the possibility that variables not included in the current model may exert additional influence on these constructs, thereby highlighting opportunities for model enhancement and theoretical refinement as advocated by Henseler et al. [97].

As summarized in **Table 4**, the model's predictive significance was evaluated using $Q^2$ values to accurately predict the sample data, as evidenced by $Q^2$ values greater than zero for all endogenous constructs. For example, 'long-term planning' has a $Q^2$ value of 0.464, indicating high predictive ability and aligning with criteria established by Geisser [98] and Stone [99]. Other constructs, such as 'agentic engagement' (AE28), showed strong predictive significance with a $Q^2$ value of 0.466. The out-of-sample predictive accuracy of the model was confirmed through the PLSpredict procedure. The Root Mean Square Error (RMSE) and Mean Absolute

**Table 3. Discriminant validity.**

|  | Long term Planning | Short term Planning | Student Engagement | Student Support |
|---|---|---|---|---|
| Long term Planning |  |  |  |  |
| Short term Planning | 0.81 |  |  |  |
| Student Engagement | 0.88 | 0.725 |  |  |
| Student Support | 0.778 | 0.677 | 0.841 |  |
| Time Attitudes | 0.528 | 0.368 | 0.592 | 0.487 |

**Table 4. MV prediction summary.**

|  | $Q^2$predict | PLS-SEM_RMSE | PLS-SEM_MAE | LM_RMSE | LM_MAE |
|---|---|---|---|---|---|
| LTP85 | 0.464 | 1.111 | 0.841 | 1.094 | 0.797 |
| LTP90 | 0.292 | 1.283 | 0.986 | 1.278 | 0.999 |
| STP83 | 0.427 | 1.18 | 0.912 | 1.181 | 0.907 |
| AE28 | 0.466 | 1.165 | 0.903 | 1.122 | 0.797 |
| AE29 | 0.415 | 1.206 | 0.935 | 1.192 | 0.872 |
| AE30 | 0.428 | 1.185 | 0.924 | 1.144 | 0.824 |
| BE17 | 0.383 | 1.164 | 0.934 | 1.139 | 0.876 |
| BE18 | 0.424 | 1.032 | 0.799 | 0.976 | 0.71 |
| CE25 | 0.24 | 0.951 | 0.674 | 0.943 | 0.685 |
| CE27 | 0.441 | 1.05 | 0.816 | 0.987 | 0.71 |
| EE20 | 0.441 | 1.18 | 0.936 | 1.129 | 0.844 |
| TA86 | 0.123 | 1.539 | 1.234 | 1.51 | 1.207 |
| TA88 | 0.175 | 1.526 | 1.223 | 1.517 | 1.201 |

*Note*: LTP: Long Term Planning; STP: Short Term Planning; AE: Agentic Engagement; BE: Behavioral Engagement; CE: Cognitive Engagement; EE: Emotional Engagement; TA: Time Attitude.

Error (MAE) for most constructs were comparable to or lower than those of the naïve benchmark model. For example, in 'long-term planning' the PLS-SEM RMSE (1.111) and MAE (0.841) were close to the corresponding Linear Model (LM) values (RMSE: 1.094; MAE: 0.797), indicating similar or slightly superior predictive performance. This demonstrates that the model is effective in fitting the current dataset and shows potential for accurate out-of-sample predictions, as suggested by Shmueli et al. [100].

To enhance the robustness of the structural equation model, control variables were judiciously added, aiming to distill the unique contributions of the primary variables of interest and to account for external influences that might confound the relationships under investigation. Incorporating control variables such as 'Gender,' which exhibited a path coefficient of 0.125, and 'Specialization,' with a coefficient of 0.143, underscored their statistically significant influence on student engagement. Incorporating these variables altered the $R^2$ value for 'Student-engagement' from 0.674 to 0.682, reflecting a slightly increased proportion of variance explained by the model when control variables were included. The $Q^2$ values (like 'Student-engagement' with a $Q^2$ value of 0.586 even after introducing controls) remained robustly positive, affirming the model's capacity to reliably predict the constructs of interest. These methodological enhancements, supported by significant path coefficients and stable $Q^2$ values, lend greater credibility to the study's findings, demonstrating the added explanatory power and nuanced insights offered by the extended model.

## Path coefficients

When the path coefficients are examined, multiple critical relationships within the controlled model emerge, as summarized in **Table 5**. 'Long-term planning' had a considerable and highly significant positive impact on 'student engagement,' with a path coefficient of 0.693 and a p-value of 0.000, showing that as students' capacity to plan for the long term improves, so does their engagement in their studies. 'Student support' had strong positive correlations with both 'long-term planning' and 'short-term planning,' with path coefficients of 0.778 and 0.678, respectively, and p-values of 0.000 for both, confirming the importance of support in improving students' planning processes. Also, 'student support' was positively associated with 'time

**Table 5. Path coefficients.**

| | Original sample (O) | Sample mean (M) | Standard deviation (STDEV) | T statistics (|O/STDEV|) | P values |
|---|---|---|---|---|---|
| LTP -> SE | 0.693 | 0.704 | 0.092 | 7.521 | 0.00 |
| STP -> SE | 0.091 | 0.083 | 0.069 | 1.312 | 0.190 |
| SS -> LTP | 0.778 | 0.778 | 0.028 | 27.648 | 0.000 |
| SS -> STP | 0.678 | 0.677 | 0.026 | 26.398 | 0.000 |
| SS -> TA | 0.490 | 0.491 | 0.042 | 11.588 | 0.000 |
| TA -> SE | 0.194 | 0.191 | 0.047 | 4.119 | 0.000 |

Note: LTP = Long term Planning; SE = Student Engagement; STP = Short term Planning; SS = Student Support; TA = Time Attitudes.

attitudes,' with a path coefficient of 0.490 and a p-value of 0.000, demonstrating that the support students receive helps shape positive attitudes about time management. This is further supported by the positive path coefficient of 'time attitudes' on 'student engagement,' which was 0.194 with a p-value of 0.000.

As summarized in **Table 6**, exploring the specific indirect effects, 'student support' was found to influence 'student engagement' through 'time-attitudes' with a coefficient of 0.095 and a statistically significant p-value of 0.000, indicating a meaningful indirect relationship. However, the indirect effect of 'student support' on 'student engagement' through 'short-term planning' was insignificant, as the p-value of 0.192 exceeds the conventional 0.05 threshold for significance. Contrastingly, 'student support' had a notable indirect effect on 'student engagement' through 'long-term planning,' with a path coefficient of 0.539 and a p-value of 0.000, signifying a strong indirect linkage between these variables.

## Hypothesis testing

Based on the results from the structural equation modeling, hypothesis testing for the study is presented as follows in Fig 4:

**H1: *Higher levels of all three forms of time efficiency predict higher levels of student engagement*.**

The evidence partially supports this hypothesis. 'Long-term planning' showed a strong positive relationship with 'student-engagement' (path coefficient = 0.693, p = 0.000), and 'time-attitudes' also positively influenced 'student-engagement' (path coefficient = 0.194, p = 0.000). However, 'short-term planning' did not significantly predict 'student engagement' (path coefficient = 0.091, p = 0.190). Therefore, while long-term planning and time attitudes are confirmed as predictors of student engagement, short-term planning was not in Tables 4 and 5.

**H2: *Higher levels of student support predict higher levels of every form of time efficiency*.**

**Table 6. Specific indirect effects.**

| | Original sample (O) | Sample mean (M) | (STDEV) | T statistics (|O/STDEV|) | P values |
|---|---|---|---|---|---|
| SS -> TA -> SE | 0.095 | 0.094 | 0.027 | 3.493 | 0.000 |
| SS -> STP -> SE | 0.062 | 0.056 | 0.047 | 1.306 | 0.192 |
| SS -> LTP -> SE | 0.539 | 0.549 | 0.085 | 6.36 | 0.000 |

Note: SS–Student Support, TA–Time Attitudes, SE–Student Engagement, STP- Short Term Planning, LTP–Long Term Planning.

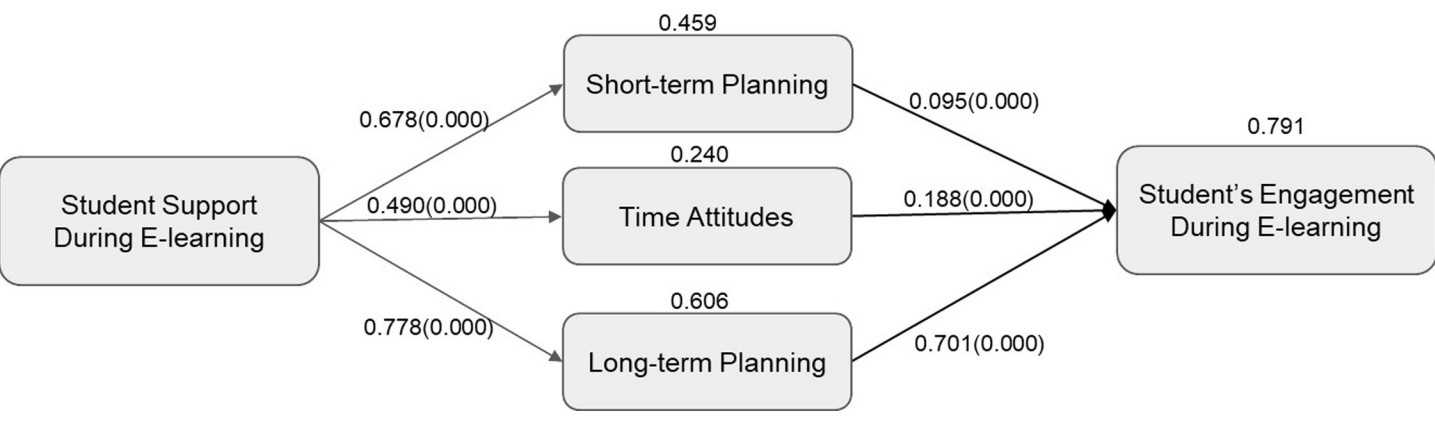

**Fig 4. Standardized path coefficients and hypothesis testing results.**

The results fully support this hypothesis. The data show that 'student support' significantly predicts 'long-term planning' (path coefficient = 0.778, p = 0.000), 'short-term planning' (path coefficient = 0.678, p = 0.000), and 'time attitudes' (path coefficient = 0.490, p = 0.000).

**H3**: *Higher levels of student support predict higher levels of all three forms of time efficiency, which in turn predicts higher levels of student engagement*.

This hypothesis is also partially supported. 'Student support' positively affects 'time-attitudes' and 'long-term planning,' which both, in turn, have a significant positive effect on 'student engagement.' The specific indirect effects through 'time-attitudes' (path coefficient = 0.095, p = 0.000) and 'long-term planning' (path coefficient = 0.539, p = 0.000) were significant. However, the indirect effect through 'short-term planning' was insignificant (path coefficient = 0.062, p = 0.192), suggesting that 'short-term planning' may not mediate between 'student support' and 'student engagement' (see Table 4 and Fig 4).

While 'student support' is confirmed as a significant predictor of all three forms of time efficiency, only 'long-term planning' and 'time attitudes' are established as mediators in the relationship between 'student support' and 'student engagement'. These findings highlight the complexity of the constructs involved and suggest a nuanced understanding of how time efficiency facets contribute to student engagement within the educational context.

## Discussion

This study considered time-efficiency variables that influence the relationship between student support and engagement during e-learning with 752 students from three Saudi universities (Taif University, Prince Sattam University, and Shaqra University). Based on engagement theory and self-determination theory, the findings partially supported the presented hypotheses. The findings are consistent with engagement theory [41] and SDT [42,43], indicating that an individual's environment largely influences their behavior. The identified trend is apparent in the observed relationships between student support, time efficiency, and engagement during e-learning. Students who feel supported are more likely to change their approaches and behaviors to attain their goals [101]. This study's findings support the concept of students' efforts to use time effectively during e-learning to achieve successful engagement and academic achievement.

Regarding the impact of time efficiency on student engagement, the analysis indicated that student engagement can be predicted by their level of time efficiency during e-learning,

consistent with published studies [30,37,86]. The findings that long-term planning and time attitudes significantly predict student engagement align with previous research showing that student engagement correlates with academic success, resilience, attendance, and retention [23,24,26,27,47] The findings indicate that short-term planning does not significantly predict student engagement during e-learning, an unanticipated finding that necessitates further study. This suggests that the nature of e-learning environments warrants long-term goal setting over short-term planning.

Regarding the impact of student support on time efficiency, my findings support the hypothesis that student support predicts all aspects of time efficiency in e-learning. The study's results revealed significant positive correlations between student support and long-term planning, short-term planning, and time attitudes. This shows that when students perceive broad support from their educational institutions (e.g., instructors, technical assistance, academic advisors, and institutional policies), they are more likely to use their time efficiently during e-learning. These findings are consistent with recent research on social and institutional support for students, which has found favorable effects on student achievement and engagement [73,76,78,79]. Previous studies have found links between teacher support and student involvement [102,103], as well as social support in general (including parental and peer support) as a predictor of student engagement [104]. The finding that long-term planning and time attitudes act as significant mediators in the relationship between student support and involvement in e-learning highlights the complicated relationship between institutional support, students' time management skills, and overall participation in e-learning environments.

## Limitations and future studies

Due to the non-normality of the data, I used PLS-SEM to accommodate complicated models with several components and indicators, resulting in strong and trustworthy prediction models despite departures from normalcy. Control variables, notably 'Gender' and 'Specialization,' with statistically significant effects on student involvement marginally boosted my model's explanatory power, highlighting the importance of demographic characteristics in understanding student involvement in e-learning situations. Limitations of this study include potential response accuracy issues due to online data collection, particularly via smartphones [105], which future studies could address through traditional methods or computer-based survey completion [106]. Online data collection can lead to self-selection bias, as only motivated students with internet access may participate, potentially skewing results. Future studies should explore ways to reduce this bias, such as combining online surveys with face-to-face or hybrid methods. Furthermore, the sample's limitation to specific Saudi universities and undergraduate demographics may restrict generalizability, necessitating more diverse sampling in future research to identify broader trends. Future studies should include a broader range of academic institutions to better capture variations in e-learning engagement and the support provided by institutions across different settings.

While the study highlights how relevant institutional support is in e-learning, it lacks intervention-based assessments, which could be a focus for future studies to evaluate the effectiveness of various support mechanisms on student outcomes. Future research could conduct studies to evaluate targeted interventions, like customized time-management training or better technical support, and their effects on student engagement and time efficiency. Also, the unexpected lack of relationship between short-term planning and engagement warrants further investigation to better understand this outcome in e-learning environments. Researchers could explore if cultural or contextual factors influence the findings or if different measures of short-term planning yield varied outcomes. Qualitative studies might provide insights into

how students engage with short-term planning in e-learning. Since this study relied on self-reported measures, common method bias may have been introduced. Future research could address this by incorporating multi-method approaches, like observational data or teacher assessments, to validate the findings and reduce bias.

## Conclusion

This study found that institutional support and students' time-efficiency skills, specifically long-term planning and time attitudes, significantly impact successful e-learning engagement. The findings have significant implications for higher education institutions continually incorporating e-learning into their curriculum, given the strong link between institutional support and time efficiency. Colleges can prioritize comprehensive support systems such as technical assistance, academic advising, and explicit course planning to help students better manage their time, leading to increased engagement in e-learning environments. The findings add to the broader theme of educational reform by advancing the case for time management in digital learning situations. Understanding these relationships is becoming increasingly crucial as e-learning becomes increasingly essential for accessible education for students geographically detached from the traditional classroom. Prioritizing the development of tailored support systems to improve students' time-efficiency abilities in long-term planning should help ensure sustainable and effective e-learning and guarantee more equitable and accessible educational opportunities.

## Author Contributions

**Conceptualization:** Tarik Abdulkrem Alwerthan.

**Data curation:** Tarik Abdulkrem Alwerthan.

**Formal analysis:** Tarik Abdulkrem Alwerthan.

**Funding acquisition:** Tarik Abdulkrem Alwerthan.

**Investigation:** Tarik Abdulkrem Alwerthan.

**Methodology:** Tarik Abdulkrem Alwerthan.

**Project administration:** Tarik Abdulkrem Alwerthan.

**Resources:** Tarik Abdulkrem Alwerthan.

**Software:** Tarik Abdulkrem Alwerthan.

**Supervision:** Tarik Abdulkrem Alwerthan.

**Validation:** Tarik Abdulkrem Alwerthan.

**Visualization:** Tarik Abdulkrem Alwerthan.

**Writing – original draft:** Tarik Abdulkrem Alwerthan.

**Writing – review & editing:** Tarik Abdulkrem Alwerthan.

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
