## [Decision Letter · Decision Letter 0]

17 Sep 2024

PONE-D-24-21575Time Efficiency as a Mediator Between Institutional Support and Higher Education Student Engagement During E-LearningPLOS ONE

Dear Dr. Alwerthan,

Thank you for submitting your manuscript to PLOS ONE. After careful consideration, we feel that it has merit but does not fully meet PLOS ONE’s publication criteria as it currently stands. Therefore, we invite you to submit a revised version of the manuscript that addresses the points raised during the review process.

We look forward to receiving your revised manuscript.

Kind regards,

Hosam Al-Samarraie

Academic Editor

PLOS ONE

Journal Requirements:

plos.org/plosone/s/file%3fid=wjVg/PLOSOne_formatting_sample_main_body.pdf%20and">When submitting your revision, we need you to address these additional requirements.

2. Thank you for stating the following financial disclosure: “The researcher would like to acknowledge Deanship of Graduate Studies and Scientific Research, Taif University for funding this work.”

3. In the online submission form, you indicated that “The data presented in this study are available on request from the author.”

All PLOS journals now require all data underlying the findings described in their manuscript to be freely available to other researchers, either 1. In a public repository, 2. Within the manuscript itself, or 3. Uploaded as supplementary information. This policy applies to all data except where public deposition would breach compliance with the protocol approved by your research ethics board. If your data cannot be made publicly available for ethical or legal reasons (e.g., public availability would compromise patient privacy), please explain your reasons on resubmission and your exemption request will be escalated for approval.

Reviewers' comments:

Reviewer's Responses to Questions

**Comments to the Author**

1. Is the manuscript technically sound, and do the data support the conclusions?

Reviewer #1: Partly

Reviewer #2: Yes

2. Has the statistical analysis been performed appropriately and rigorously? 

Reviewer #1: Yes

Reviewer #2: I Don't Know

3. Have the authors made all data underlying the findings in their manuscript fully available?

Reviewer #1: Yes

Reviewer #2: Yes

4. Is the manuscript presented in an intelligible fashion and written in standard English?

Reviewer #1: No

Reviewer #2: Yes

5. Review Comments to the Author

Reviewer #1: Review Comments

Time Efficiency as a Mediator between Institutional Support and Higher Education Student Engagement During E-Learning

Thank you for the opportunity to review this manuscript. The subject matter is important for the field of discipline. I have however, indicated some areas that need to be improved to enhance the study.

Abstract:

- The first sentence is too long making it difficult to connect with the proposed linkages among the variable factors.

- “This study employed a cross-section research design” is fine. Consider deleting questionnaire-based.

Introduction

- Lines 3-4: e-learning has existed over several decades. The researcher’s attempt to link it with the “many radical

changes in the past five years as a result of technological development and globalization of knowledge” makes it non-

factual.

- Line 7: establish a good linkage between this sentence and the previous one to aid cohesion and comprehension.

Since “Though programs …” seem to be completely out of place.

- Line 12: first time mention of UNESCP should be in full and subsequently abbreviated version can be used. Moreover,

is the UNESCP education similar to eLearning?

- Paragraph 2, kindly check and correct grammar in the sentence one in order to make the meaning much clearer.

- the gap in knowledge that the study is seeking to address is not clear. Authors must make the gap clear and indicate

why it is important for this research to address it.

Literature Review

Line 1: revise as follows: “ the concept of time-efficiency has been studied across various fields” such as … indicate

the examples of the various fields you are referring to.

- the justification for the mediating role of time-efficiency is lacking.

- The abbreviation of Engagement theory (ET) in the sub-heading should be in bracket.

- Generally, the manuscript requires major proof-reading and editing to correct certain errors seen in the work.

Attention should be paid to proper punctuation, very long paragraphs, long sentences (pg.8) repetition of certain

phrases eg. Authority figures, etc

- The justification for the use of engagement theory is missing from the discussion. It will do the manuscript a great

deal of good if authors consider this suggestion. The theory was hardly discussed except the first sentence in the last paragraph on pg. 6.

- There is lack of linkages between the various paragraphs making the manuscript lack coherence.

- The first use of SDT should be in full.

- The literature review should be able to provide a critical review of existing researches. Going this route could

minimize the many citations considerably.

- The generation of the research gap justifying the stated hypotheses is still missing.

correct to Study hypotheses

- H1 -3: There was no discussion on the three forms of time-efficiency in the LR. Authors may consider addressing this

since that will provide the bases for their presence in the study.

- Moreover, what is the relevance of “Figure 1, 2, 3” ending the hypothetical statements? If it is for reference purposes,

the consider placing them in a bracket. For example, (see Figure 1)

Methods

- Methodology should be revised considerably to reflect the style of a manuscript.

- Was there an estimated population, and sample size? What sampling technique (s) was/were used?

- Consider including the source of the instrument used to measure student support.

Results

- How was the non-normal nature of the data treated?

- The write-up on “Data Analysis” appears overly long. It should be revised to reflect the style of a manuscript.

Path coefficients

- Did the study control for some variables? If so kindly indicate them.

- The results need to be reorganized in such a way that clearly depicts what is demonstrated in the tables. As at now

the authors make reference to results that is non-existent in tables. For example, Lines 1 – 4, authors make

reference to a substantial and highly significant positive impact on student engagement with path coefficient of 0.693

and a p-value of 0.00 in table 4. However, there is no such results in table 4. The study will benefit from a thorough

review of the results.

Discussion

The discussion should be revised to suit the style of a manuscript.

References

- Generally, the references are too many and a careful revision of the entire manuscript could reduce it considerably.

- In-text citation and references could benefit from a thorough revision to conform with the journal’s standard. For

example, pgs. 5, 6,7,8 and many more throughout the study.

Reviewer #2: The study makes significant contributions to the understanding of time-efficiency in e-learning environments, specifically within the context of Saudi Arabia. The author has effectively addressed a gap in the literature, offering well-supported arguments that answer the research questions.

Strengths:

- The study is well-grounded in theoretical frameworks providing a solid conceptual basis.

- The use of a large sample across multiple Saudi universities enhances the study's credibility.

- Visual data representations are clear and facilitate a better understanding of the relationships between variables.

- The writing is clear, concise, and well-structured, making the study accessible to readers.

Recommendations:

- Provide more clarity in defining types of student support considered in the study.

- Expand on the Saudi Arabian context in relation to e-learning. This will help frame the study's findings more effectively for an international audience.

- Offer specific examples of the obstacles students faced during e-learning (Page 3, Paragraph 2).

- The first two sentences discussing time-efficiency on Page 5, Paragraph 1 lack citations.

- At the end of each section in the literature review, consider summarising how this study fills the identified gaps.

- Include a table summarising demographics details of the participants (e.g., age, university year).

- Provide a more detailed explanation of why short-term planning was found to be less significant.

- Since SDT and ET are not frequently mentioned in the study, the use of abbreviations might be confusing.

Minor Corrections:

- Avoid repeating the author's name twice in the same sentence. For example, instead of "Adams and Blair (Adams & Blair, 2019a)," cite the authors once.

- Correct the misspelling of UNESCO on Page 3, Paragraph 1, and ensure the inclusion of a citation.

- There are doubled parentheses on Page 5 and Page 21.

- Replace the ampersand (&) used in citations with "and," and avoid overusing "e.g." before citations to maintain the academic style.

6. PLOS authors have the option to publish the peer review history of their article (what does this mean?). If published, this will include your full peer review and any attached files.

Reviewer #1: No

Reviewer #2: **Yes: **Eshrag Aljehani

---

## [Author Response · Author response to Decision Letter 0]

22 Oct 2024

Dear Editor and Reviewers,

We sincerely appreciate your insightful and constructive feedback on our manuscript. We have carefully incorporated your comments and made significant revisions to improve the clarity, organization, and overall quality of the paper. In addition, the manuscript has been reorganized, and corrections were made to address certain language issues. We are grateful for your efforts in strengthening this manuscript. Below, we detail how we have addressed your concerns.

Editor

We have included the appropriate statement regarding the funder's role in the cover letter.

Additionally, we have submitted the data as a supplementary file.

Reviewer 1

Abstract

We updated the first sentence for clarity and removed the term "questionnaire-based" as advised.

Introduction

We updated the statement regarding e-learning to better reflect its historical context and recent advances.

We increased sentence linking to enhance cohesiveness and comprehension.

We spelled out "UNESCO" in full at the first mention.

We clarified the grammar of the second paragraph and clearly stated the gap in knowledge as a closing paragraph of the literature review.

Literature Review

We modified the opening statement to include examples of fields where time efficiency has been explored.

We presented an enhanced argument for time efficiency's mediating role.

We updated the abbreviation format for Engagement Theory and eliminated the use of the acronym.

We proofread and edited the manuscript, correcting punctuation, paragraph length, and redundancy.

We expanded the concept of Engagement Theory to better support its application in our study and adjusted paragraph linkages to improve coherence.

We defined Self-Determination Theory (SDT) in full and eliminated the acronym's usage.

We revisited the literature review and reduced unnecessary citations, highlighting the research gap that supports our hypotheses.

Hypotheses

We changed the heading to "Study Hypotheses" and included a discussion of the three types of time efficiency in the Literature Review for context.

We updated the format of figure references to "(see Figure)".

Methods

We updated the methodology section to better align with the style of a manuscript.

We included details about the estimated population, sample size, and sampling method used, as well as the source of the instrument to assess student support.

Non-normal Nature of Data

We expanded the Data Analysis section to address how we treated the non-normal nature of our data and explained our shift from covariance-based SEM to Partial Least Squares SEM (PLS-SEM).

Data Analysis Write-up

We substantially revised and condensed the Data Analysis section to better match the manuscript's style, focusing on key methodological decisions and their rationale.

Control Variables

We added information about the control variables used, specifically noting that we included "Gender" and "Specialization" and reported their effects on student engagement.

Results Reorganization

We thoroughly revised the Results section to ensure clear alignment between the text and the tables, correcting discrepancies, such as the one you pointed out regarding the path coefficient and p-value for the relationship between long-term planning and student engagement.

Discussion Revision

We revised the Discussion section to better interpret our findings within the context of existing literature and theoretical frameworks.

References

We reduced the number of references and edited the in-text citations and reference list to meet the journal's standards.

Reviewer 2

Short-term Planning Significance

We expanded our discussion on the significance of short-term planning, providing explanations for its limited impact in our study and suggesting areas for future research in e-learning contexts.

Contextual and Support Definitions

We provided specific definitions for the types of student support evaluated in our study.

We elaborated on the Saudi Arabian context of e-learning to better explain our findings, including examples of challenges students faced.

We added citations to support the discussion on time efficiency and included a paragraph at the end of the literature review to demonstrate how our study fills identified gaps.

We also added a table summarizing participants' demographics and offered a detailed explanation for the reduced significance of short-term planning in our findings.

Abbreviations

We spelled out Self-Determination Theory (SDT) and Engagement Theory (ET) and used their full terms consistently throughout the manuscript.

We believe these revisions have significantly improved the clarity, organization, and overall quality of our manuscript. We appreciate your valuable input and hope that our responses and revisions adequately address your concerns.

We are looking forward to hearing from you.

Best regards,

Authors

---

## [Decision Letter · Decision Letter 1]

12 Nov 2024

PONE-D-24-21575R1Time Efficiency as a Mediator Between Institutional Support and Higher Education Student Engagement During e-LearningPLOS ONE

Dear Dr. Alwerthan,

Thank you for submitting your manuscript to PLOS ONE. After careful consideration, we feel that it has merit but does not fully meet PLOS ONE’s publication criteria as it currently stands. Therefore, we invite you to submit a revised version of the manuscript that addresses the points raised during the review process.

We look forward to receiving your revised manuscript.

Kind regards,

Hosam Al-Samarraie

Academic Editor

PLOS ONE

Journal Requirements:

Reviewers' comments:

Reviewer's Responses to Questions

**Comments to the Author**

1. If the authors have adequately addressed your comments raised in a previous round of review and you feel that this manuscript is now acceptable for publication, you may indicate that here to bypass the “Comments to the Author” section, enter your conflict of interest statement in the “Confidential to Editor” section, and submit your "Accept" recommendation.

Reviewer #2: (No Response)

2. Is the manuscript technically sound, and do the data support the conclusions?

Reviewer #2: Yes

3. Has the statistical analysis been performed appropriately and rigorously? 

Reviewer #2: I Don't Know

4. Have the authors made all data underlying the findings in their manuscript fully available?

Reviewer #2: Yes

5. Is the manuscript presented in an intelligible fashion and written in standard English?

Reviewer #2: No

6. Review Comments to the Author

Reviewer #2: The author has made improvements in addressing the feedback, but there are still minor issues related to grammar, typos, flow, and readability. These adjustments will further enhance the clarity and quality of the manuscript. Below are some examples of the corrections needed:

- Page 2 > Line 4: There is a repetition of the word "in".

- Some sentences are a bit long and could be broken down into shorter sentences to improve readability. For instance, the sentence on Page 3 > Paragraph 1 > last sentence could be simplified for easier comprehension.

- Page 3 > Paragraph 1 > Line 8: The term "pandemic" is mentioned twice.

- Page 4 > Paragraph 3: The phrase "individuals who" is repeated.

- Page 5, Paragraph 1: Check for repeated punctuation.

- Ensure the citation style is consistent throughout the document.

- It would be beneficial to add a paragraph to conclusion that discusses the limitations of the study. This paragraph could also include recommendations for future research.

7. PLOS authors have the option to publish the peer review history of their article (what does this mean?). If published, this will include your full peer review and any attached files.

Reviewer #2: **Yes: **Eshrag Aljehani

---

## [Author Response · Author response to Decision Letter 1]

21 Nov 2024

Dear Reviewer

I hope this message finds you well. I am writing to express my sincere gratitude for your insightful comments and valuable feedback on my manuscript. Your comments have significantly contributed to improving the quality and clarity of the work.

I have carefully addressed all your comments and suggestions, as detailed in the revised manuscript. The tracked changes file highlights the modifications made. These changes include, but are not limited to the following:

Reviewer Comments and Author Responses

Grammar, Typos, Flow, and Readability:

The manuscript was thoroughly proofread to resolve all issues, including:

Correcting the repetition of 'in' on Page 2, Line 4.

Simplifying the long sentence in Page 3, Paragraph 1, last sentence.

Removing the duplicate mention of 'pandemic' on Page 3, Paragraph 1, Line 8.

Eliminating the repeated phrase 'individuals who' on Page 4, Paragraph 3.

Resolving repeated punctuation on Page 5, Paragraph 1.

Citation Style Consistency:

Ensured uniform citation formatting throughout the manuscript.

Discussion on Limitations and Future Research:

Added a new 'Limitations and Future Studies' section before the conclusion, addressing key areas such as data collection limitations, potential biases, and suggestions for future research. This includes specific directions on demographic influences, intervention-based assessments, and employing multi-method approaches to validate findings.

Once again, thank you for your constructive comments, which I believe have greatly enhanced the manuscript. Please feel free to let me know if there are any additional aspects that require further attention.

I look forward to your feedback.

Best regards,

---

## [Editor Report · Decision Letter 2]

26 Nov 2024

Time Efficiency as a Mediator Between Institutional Support and Higher Education Student Engagement During e-Learning

PONE-D-24-21575R2

Dear Dr. Alwerthan,

We’re pleased to inform you that your manuscript has been judged scientifically suitable for publication and will be formally accepted for publication once it meets all outstanding technical requirements.

Kind regards,

Hosam Al-Samarraie

Academic Editor

PLOS ONE

Additional Editor Comments (optional):

Thank you for addressing all comments.
---

## [Editor Report · Acceptance letter]

28 Nov 2024

PONE-D-24-21575R2 

PLOS ONE

Dear Dr. Alwerthan, 

I'm pleased to inform you that your manuscript has been deemed suitable for publication in PLOS ONE. Congratulations! Your manuscript is now being handed over to our production team.

Kind regards, 

on behalf of

Dr Hosam Al-Samarraie 

Academic Editor

PLOS ONE